# Distribution of HPV Types in Tumor Tissue from Non-Vaccinated Women with Cervical Cancer in Norway

**Sveinung Wergeland Sørbye [1],\* , Bente Marie Falang [2] and Mona Antonsen [1]**

[1] Department of Clinical Pathology, University Hospital of North Norway, 9038 Tromsø, Norway; mona.antonsen@unn.no
[2] PreTect AS, 3490 Klokkarstua, Norway; bente.falang@pretect.no
\* Correspondence: sveinung.wergeland.sorbye@unn.no; Tel.: +47-77-62-72-23

**Abstract:** Background: Understanding the distribution of HPV types in cervical cancer cases is crucial for evaluating the effectiveness of HPV screening and vaccination in reducing cervical cancer burden. This study aimed to assess genotype prevalence in the pre-vaccine era among 178 cervical cancer cases detected during a 20-year screening period in Northern Norway and compare the potential efficacy of HPV vaccines in preventing cervical cancer. Methods: A total of 181 formalin-fixed paraffin-embedded (FFPE) tissue samples from non-vaccinated women diagnosed with cervical cancer between 1995 and 2015 in Troms and Finnmark, Norway, were analyzed using a 45-type HPV DNA test. The results were compared to a 7-type HPV mRNA test targeting oncogenic types included in the nonavalent HPV vaccine. Results: Invalid HPV test results were observed in 1.7% (3/181) of the samples and were subsequently excluded from further analysis. Among the remaining cases, 92.7% (165/178) tested positive for HPV using any test combination. HPV DNA was detected in 159 cases (89.3%), while HPV mRNA was detected in 149 cases (83.7%). The most prevalent HPV types were 16 and 18, responsible for 70.8% of the cases, with the nonavalent vaccine types accounting for 86.6% of cases. HPV 35 was identified in eight cases (4.5%). Conclusion: The bivalent/quadrivalent HPV vaccines have the potential to prevent 76.4% (126/165) of HPV-positive cervical cancer cases, while the nonavalent vaccine could prevent 93.3% (154/165) of cases. Tailoring screening strategies to target HPV types with the highest oncogenic potential may improve cervical cancer detection and enable targeted interventions for high-risk individuals. The use of a 7-type HPV mRNA test holds promise as an advantageous approach.

**Keywords:** cervical cancer; HPV screening; mRNA; genotype distribution; vaccine efficacy





## 1. Introduction

Human papillomavirus (HPV) has been unequivocally established as the leading cause of cervical cancer, driven by the overexpression of viral oncoproteins E6 and E7 [1]. Despite substantial evidence demonstrating the effectiveness of national screening and vaccination programs in preventing and treating cervical cancer, it remains the fourth most prevalent cancer among women worldwide. In 2018, the World Health Organization (WHO) launched an ambitious program aimed at eliminating cervical cancer, targeting an incidence rate of less than 4 per 100,000 women in all countries [2]. However, low- and middle-income countries bear the highest burden of cervical cancer, and the success of their screening and vaccination efforts relies on the global political commitment to achieve this ambitious goal. The WHO's elimination strategy follows a "90-70-90" approach, emphasizing high percentages of vaccination uptake, screening participation, and treatment accessibility [2]. Nonetheless, the implementation of these strategies varies across regions due to differences in available healthcare resources, resulting in disparities in screening and vaccination rates influenced by health policies, personal barriers, and structural impediments [3].

In Norway, a national cervical cancer screening programme (NCCSP) has been in place since 1995, recommending cervical cytology every three years for women aged

25–69 years. In 2015, the program transitioned to primary HPV screening every five years for women aged 34–69 years, which was further expanded in 2023 to include all women aged 25–69 years due to the solid scientific evidence that HPV testing is more sensitive and safer [4]. The current coverage of the NCCSP in Norway is reported to be 70%, which is in line with the WHO target [5]. The HPV vaccine has been included in the Norwegian childhood immunization program since 2009, providing the quadrivalent (HPV16/18/6/11) or bivalent (HPV16/18) HPV vaccine to 12-year-old girls and to boys starting in 2018. Despite having one of the highest vaccine uptakes globally, at 92% [6], Norway's cervical cancer incident rate of 10.8/100,000 is still considerably higher than the goal of elimination [7].

Since Zur Hausen's groundbreaking discovery linking HPV to cervical cancer [8], HPV types 16 and 18 have been recognized as the predominant types, accounting for 70% of all cervical cancer cases. As a result, the first generation of HPV vaccines focused on protecting against these types. Building on decades of research that has identified the varying cancer-causing potentials of different HPV genotypes [9–14], the nonavalent vaccine was developed to target the additional types 31, 33, 45, 52, and 58, which provides up to 90% protection against cervical cancer [15]. Importantly, type-specific HPV prevalence varies across the severity of cervical lesions. Worldwide HPV statistics gathered by the Institute Catalan Oncology (ICO) and the International Agency for Research on Cancer (IARC) inform of the top 10 HPV types identified among women with and without cervical lesions, confirming the seven oncogenic types included in the nonavalent vaccine as the most prevalent in cervical cancer, illustrated in Figure 1 [16].

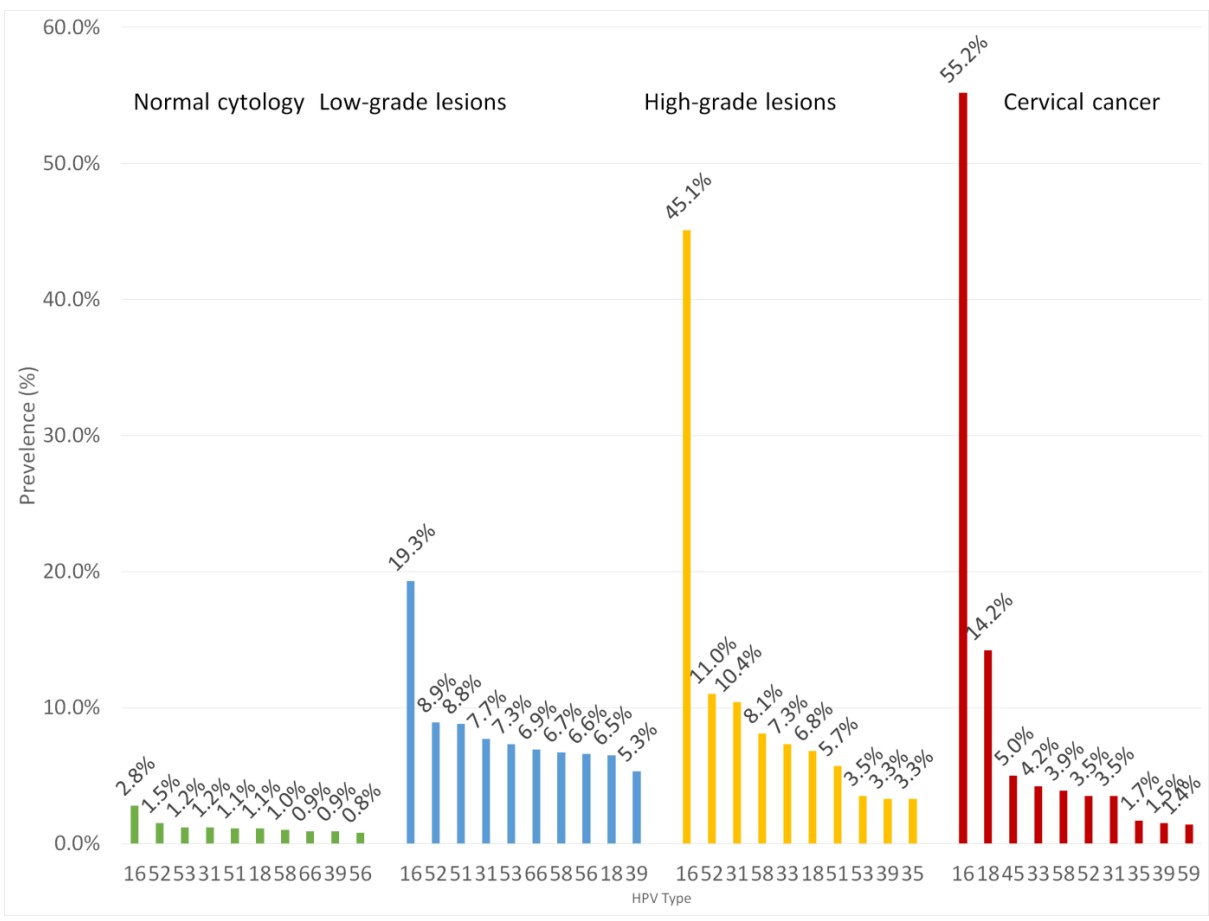

**Figure 1.** Comparison of the ten most frequent HPV oncogenic types worldwide among women with and without cervical lesions. Adapted from: World Statistics, Full report, Figure 78 [16].

For obvious reasons, few studies have been able to use cervical cancer as a histological endpoint to compare the performance of different HPV tests; thus, detection of cervical intraepithelial neoplasia grade 2/3 (CIN2/3+) has been used as a surrogate measure. However, understanding the distribution of HPV types among confirmed cervical cancer cases is critical for assessing the effectiveness of HPV screening and vaccination in reducing the burden of cervical cancer.

In this study, we aimed to determine the prevalence and distribution of HPV types in cervical cancer cases during a 20-year period preceding systematic HPV vaccination in Norway. Contributing to the understanding of the natural distribution of HPV in cervical cancer cases in a pre-vaccination era, the data might serve as a relevant baseline for future studies evaluating the impact of vaccination on HPV type distribution in the post-vaccine era. We also sought to compare the efficacy of the bivalent, quadrivalent, and nonavalent HPV vaccines in preventing cervical cancer. To add knowledge to the existing gap in the literature on how HPV mRNA tests perform in cervical cancer tissue samples, a 7-type HPV mRNA test was included and compared to the 45-type HPV-DNA test used for genotyping.

## 2. Materials and Methods

### 2.1. Study Design

This was a retrospective case-series design, analyzing archived formalin-fixed paraffin-embedded (FFPE) tissue samples from non-vaccinated women diagnosed with cervical cancer between 1995 and 2015 in Troms and Finnmark counties in Northern Norway. Material for HPV testing was provided by the clinical biobank of the Department of Clinical Pathology, University Hospital of North Norway. In April 2018, archived tissue samples were tested by a 45-type HPV DNA- and a 7-type mRNA test to evaluate the HPV-type distribution in confirmed cervical cancer cases, pre-vaccination.

### 2.2. Study Population

From 1995 to 2015, 181 women attending cervical cancer screening in Troms and Finnmark were diagnosed with cervical cancer, being eligible for retrospective analyses. The women's age at the time of diagnosis ranged from 24 to 93 years old, being born between 1909 and 1989. None of the women in the study cohort had been HPV-vaccinated nor born in cohorts that had received the vaccine. Tumor tissue was collected from cancer-diagnosed women who had undergone a biopsy, conization, or hysterectomy in line with Norwegian guidelines for follow-up of abnormal cervical cancer screening results. All cervical cancer cases were diagnosed by two experienced pathologists at the Department of Clinical Pathology at the University Hospital of North Norway. To ensure the representativeness of the tumor tissue, a second review was conducted in 2018 prior to study enrolment. Any tumor tissue sample not being confirmed as a histological representative and/or samples reported as invalid by any of the HPV tests were excluded. After exclusions, 178 women were included in the final study population. A total of 39.9% (71/178) of cervical cancer cases were detected through routine screening, whilst 107 cases were diagnosed in women seeking medical attention due to symptoms.

### 2.3. Pretreatment, Isolation, and Purification of Nucleic Acids

To minimize the risk of material transfer during tissue specimen sectioning, a thorough decontamination was performed between every FFPE sample sectioned. Equipment and work surfaces were wiped with alcohol, in addition to sectioning an empty paraffin block between every FFPE tissue sample processed. Tissue thickness was standardized at 10 μm with a maximal input of 50 μm for the isolation procedure. Following an optimized research protocol to free nucleic acids from the FFPE tissue samples, the replacement of the wax with water was done through a series of soaks in Xylene followed by dilutions of Ethanol (99-90-70%). Digestion was performed using Proteinase K for 1 h at 60 °C, and removal of formalin-induced crosslinks was achieved by heating to 100 °C for 30 min., followed by incubation at room temperature for 15 min. Dewaxed, digested, and de-crosslinked

samples were subsequently lysed in a high molar lysis buffer containing chaotropic salt, and nucleic acids were isolated using the commercially available extraction kit (PreTect X, Klokkarstua, Norway) according to the manufacturer's instructions. Eluted nucleic acids were kept at −70 °C prior to DNA/mRNA testing. Testing was performed on the same eluate for each patient. All HPV test results were blinded for laboratory personnel performing the testing.

*2.4. Human Papillomavirus Testing*

HPV-DNA genotyping was performed using the Reverse Line Blot (RLB) assay on the Broad-Spectrum General Primers 5+/6+ polymerase chain reaction (PCR) products [17,18]. The assay reports 39 individual types (HPV 6, 11, 16,18, 26, 30, 31, 33, 34, 35, 39, 40, 42, 43, 44, 45, 51, 52, 53, 54,55, 56, 57, 58, 59, 61, 64, 66, 67, 68, 69, 70, 71, 72, 73, 81, 82/MM4, 82/IS39, and CP6108) in addition to 6 rare HPV types (HPV32, 83, 84, 85, 86, and JC9710) reported as a pool. All HPV-negative samples reported by RLB were controlled for the presence of human B-globin by PCR, and if not detected, the sample was excluded from the study analysis.

HPV mRNA E6/E7 expression was qualitatively reported by the PreTect HPV-Proofer '7 test (PreTect AS, Klokkarstua, Norway), genotyping HPV mRNA 16, 18, 31, 33, 45, 52 and 58. Synthetic positive and negative controls for all individual targets monitored the entire test process for final validation of the results, as reported by the PreTect Analysis Software. The amplification was carried out in a 20 µL reaction volume that consisted of 5 µL nucleic acid template solution, 10 µL Primer-Beacon Reagent Mix and 5 µL Enzyme Solution. An intrinsic sample control (ISC) detecting mRNA of a housekeeping gene (Glyceraldehyde-3-phosphate dehydrogenase, GAPDH) assessed specimen quality, and if invalid, the sample was excluded from the study analysis.

*2.5. Study Outcomes*

The aim of this study was to assess the natural HPV genotype distribution among cervical cancer cases diagnosed in non-vaccinated women in North Norway over two decades and to evaluate how a 7-type HPV mRNA test performed in cancer tissue samples compared to an HPV-DNA-based test. Based on the identified causative HPV genotype in the tumor tissue samples, the possible efficacies to prevent cervical cancer of the bivalent, quadrivalent, and nonavalent HPV vaccines were calculated.

Statistical Analysis: Data were analyzed using Statistical Package for Social Sciences (SPSS) v28 (IBM). For comparisons between the 45-type HPV DNA test and the 7-type HPV mRNA test, we used the McNemar test with continuity correction, which is appropriate for analyzing paired nominal data in a 2 × 2 table. A *p*-value < 0.05 was considered as the significance level for all statistical tests. Additionally, we calculated Cohen's kappa values to assess the agreement between the two different HPV tests. Kappa values were used to evaluate the level of agreement beyond chance between the two tests.

Ethical approval: The protocol for this study was approved as a quality assurance study by the Regional Committee for Medical and Health Research Ethics (REK Nord 2016/1333). Norwegian regulations exempt quality assurance studies from obtaining written informed consent from patients.

**3. Results**

Out of the 181 FFPE tissue samples analyzed, three samples (1.7%) had invalid HPV test results and were excluded from the analysis. Among the 178 valid cases, 159 (89.3%) tested positive for the 45-type HPV DNA test, and 149 (83.7%) tested positive for the 7-type HPV mRNA test (Table 1).

**Table 1.** HPV genotype prevalence reported by HPV DNA and mRNA tests.

| | HPV DNA Results | | | HPV mRNA Results | | |
|---|---|---|---|---|---|---|
| Type | Frequency n | Percent (%) | Cumulative Percent (%) | Frequency n | Percent (%) | Cumulative Percent (%) |
| HPV 16 | 89 | 50.0 | 50.0 | 90 | 50.6 | 50.6 |
| HPV 18 | 31 | 17.4 | 67.4 | 32 | 18.0 | 68.6 |
| HPV 45 | 13 | 7.3 | 74.7 | 13 | 7.3 | 75.9 |
| HPV 33 | 9 | 5.1 | 79.8 | 8 | 4.5 | 80.4 |
| HPV 35 | 8 | 4.5 | 84.3 | * | * | * |
| HPV 31 | 4 | 2.2 | 86.5 | 4 | 2.2 | 82.6 |
| HPV 39 | 2 | 1.1 | 87.6 | * | * | * |
| HPV 52 | 1 | 0.6 | 88.2 | 1 | 0.6 | 83.2 |
| HPV 58 | 1 | 0.6 | 88.8 | 1 | 0.6 | 83.8 |
| HPV 73 | 1 | 0.6 | 89.4 | * | * | * |
| Negative | 19 | 10.7 | 100.0 | 29 | 16.3 | 100.0 |
| Total | 178 | 100.0 | | 178 | 100.0 | |

* HPV type not included in the 7-type HPV mRNA test.

Both tests had concordant positive results for 146 cases, whilst 16 cases were reported as negative for both DNA and mRNA. The overall concordance rate for the two different HPV tests was 91.0% (162/178), with a kappa value of 0.617 and *p*-value < 0.001. The McNemar chi-squared statistic was equal to 5.06 (*p* = 0.024), as presented in Table 2.

**Table 2.** HPV DNA x HPV mRNA crosstabulation.

| | | HPV mRNA | | Total |
|---|---|---|---|---|
| | | Negative | Positive | |
| **HPV DNA** | Negative | 16 | 3 | 19 |
| | Positive | 13 | 146 | 159 |
| | Total | 29 | 149 | 178 |

Among the 159 women with positive HPV DNA results, 146 (91.8%) had co-existent overexpression of mRNA E6/E7 from the seven types included in the mRNA test. The most prevalent HPV types among the 149 women with positive HPV mRNA test results were HPV 16 (60.4%, 90/149), HPV 18 (21.5%, 32/149), and HPV 45 (8.7%, 13/149).

Of the 13 cervical cancer cases with a positive HPV DNA test but negative mRNA results, eight cases were genotype HPV 35, two cases were HPV 39, and the remaining three cases were identified as one of each of the HPV types 16, 33, and 73. Three cases had positive mRNA test results (two HPV 16, one HPV 18) whilst reported as HPV DNA negative.

A total of 16 cervical cancer cases were reported as HPV DNA and mRNA negative. Reviewing available screening history for the presumably false HPV negative biopsies, 3/16 had a positive HPV 16 result as their last test result analyzing the liquid-based cytology (LBC) specimen. A total of 13 patients had never been HPV-tested prior to their cancer diagnosis (Table 3).

In total, 92.7% (165/178) of the study participants had a positive HPV test result by any test combination; 146 HPV DNA+/mRNA+, 13 HPV DNA+/mRNA−, 3 HPV DNA-/mRNA+ and 3 HPV DNA−/mRNA− but HPV positive LBC sample. Of the 71 cervical cancers detected through routine screening, 60 (84.5%) cases were caused by the predominant genotypes HPV 16 and 18. In 93.0% (66/71) of the cases, the causative HPV type was one of the seven hrHPV types covered by the nonavalent HPV vaccine (HPV16, 18, 31, 33, 45, 52, 58). Among the 107 cervical cancer cases detected in women with clinical symptoms prior to diagnosis, 61.7% (66/107) were induced by HPV types 16 and 18, whilst 82.2% (88/107) were contributed by the seven hrHPV types, respectively (Table 4).

**Table 3.** Screening characteristics of HPV negative FFPE cervical cancer tissue samples.

| Sample | Biopsy (Year) | Age CxCa Diagnosis | Routine Screening (YES/NO) | Last Normal Cytology (Years) | First Abnormal Cytology (Years) | Last Cytology Diagnosis | Screening Failure (YES/NO) | HPV Test LBC (NT/Type) |
|---|---|---|---|---|---|---|---|---|
| 1 | 1996 | 38 | NO | 14 | 0 | Normal | NO | NT |
| 2 | 1996 | 34 | YES | 1 | 0 | ASC-H | YES | 16 |
| 3 | 1998 | 45 | YES | 5 | 5 | HSIL | YES | NT |
| 4 | 1998 | 57 | NO | 1 | 5 | ASC-US | YES | NT |
| 5 | 2005 | 63 | NO | 32 | 0 | Normal | NO | NT |
| 6 | 2006 | 56 | NO | 1 | 3 | ASC-H | YES | NT |
| 7 | 2007 | 78 | NO | 6 | 28 | Normal | YES | NT |
| 8 | 2010 | 67 | YES | 5 | 5 | HSIL | NO | NT |
| 9 | 2010 | 75 | NO | 3 | 0 | Normal | YES | NT |
| 10 | 2011 | 49 | YES | 0 | 0 | ASC-H | YES | NT |
| 11 | 2011 | 70 | NO | 8 | 28 | Normal | NO | NT |
| 12 | 2012 | 47 | NO | 1 | 4 | ASC-US | YES | NT |
| 13 | 2012 | 79 | NO | 0 | 0 | Normal | YES | NT |
| 14 | 2014 | 44 | NO | 3 | 0 | ASC-H | YES | 16 |
| 15 | 2015 | 55 | NO | 9 | 0 | HSIL | NO | NT |
| 16 | 2015 | 30 | YES | Missing | 1 | LSIL | NO | 16 |

**Table 4.** HPV type distribution in cervical cancer cases, overall and by screen-detected and among women with symptoms prior to their cancer diagnosis.

| HPV Type (DNA/mRNA) | Frequency n | Percent (%) | Cumulative Percent (%) | Screen Detected n (%) | Symptom Detected n (%) |
|---|---|---|---|---|---|
| HPV 16 | 94 | 52.8 | 52.8 | 41 (57.7) | 53 (49.5) |
| HPV 18 | 32 | 18.0 | 70.8 | 19 (26.8) | 13 (12.1) |
| HPV 45 | 13 | 7.3 | 78.1 | 2 (2.8) | 11 (10.3) |
| HPV 33 | 9 | 5.1 | 83.2 | 3 (4.2) | 6 (5.6) |
| HPV 35 | 8 | 4.5 | 87.7 | 1 (1.4) | 7 (6.5) |
| HPV 31 | 4 | 2.2 | 89.9 | 1 (1.4) | 3 (2.8) |
| HPV 39 | 2 | 1.1 | 91.0 | 0 (0.0) | 2 (1.9) |
| HPV 52 | 1 | 0.6 | 91.6 | 0 (0.0) | 1 (0.9) |
| HPV 58 | 1 | 0.6 | 92.2 | 0 (0.0) | 1 (0.9) |
| HPV 73 | 1 | 0.6 | 92.8 | 1 (1.4) | 0 (0.0) |
| Negative | 13 | 7.3 | 100.0 | 3 (4.2) | 10 (9.3) |
| Total | 178 | 100.0 | | 71 (100.0) | 107 (100.0) |

Mean age at the time of cancer diagnosis varied across genotypes inducing cancer, from 44.3 years of age for HPV 18, 45.3, and 45.5 for HPV 16 and 45, respectively. Combining the other seven genotypes identified among the participants, the mean age was 51.6, whilst women with an HPV negative cancer was considerably older, at 59.9 years of age (Table 5).

**Table 5.** Age at the time of cervical cancer diagnosis by HPV type.

| HPV Type | Age Mean | N | SD | Cumulative % |
|---|---|---|---|---|
| Negative | 59.9 | 13 | 13.28 | 7.3 |
| 16 | 45.3 | 94 | 15.29 | 60.1 |
| 18 | 44.3 | 32 | 11.37 | 78.1 |
| 45 | 45.5 | 13 | 10.49 | 85.4 |
| Other * | 51.6 | 26 | 15.49 | 100.0 |
| Total | 47.1 | 178 | 14.74 | |

* Other HPV types (31, 33, 35, 39, 52, 58, 73).

## 4. Discussion

This study contributes to the existing research addressing HPV genotype prevalence and distribution in confirmed cervical cancer cases in the Nordic region, allowing calculations and comparison of the efficacy of the three licensed prophylactic HPV vaccines that are available for the prevention of cervical cancer. Furthermore, the presented results confirmed the suitability of a 7-type HPV mRNA type in FFPE tumor tissue samples.

Our findings demonstrated that 92.7% of the non-HPV vaccinated women diagnosed with cervical cancer between 1995 and 2015 in Troms and Finnmark counties tested positive for HPV in at least one test, identifying HPV 16 and 18 as the predominant types responsible for 70.8% of the cases. This aligns with global statistics where HPV 16 and 18 account for approximately 70% of all cervical cancer cases, while HPV 31, 33, 45, 52, and 58 contribute to an additional 20% of cancer incidents. This genotype distribution has been found to be consistent from a worldwide perspective [19–21], guiding the HPV vaccine improvements for maximal prevention of cervical cancer.

Based on the genotype distribution reported from the pre-vaccine era, it is evident that the implementation of the bivalent or quadrivalent HPV vaccines could potentially prevent 76.4% (126/165) of the HPV-positive cervical cancer cases among women in Troms and Finnmark county, Norway. Furthermore, the nonavalent HPV vaccine has the potential to improve prevention to 93.3% (154/165). These figures highlight the oncogenic potential associated with the seven HPV types and the significant impact of HPV vaccination in reducing the cervical cancer burden in our specific region.

Inevitably, HPV vaccination will change the HPV prevalence among the screening population. Affirming our findings, a recent population-based HPV prevalence study conducted in the Nordic region by Nygård et al., suggested that "HPV screening tests in the post-vaccination era might perform better if restricted to the HPV types in the nonavalent vaccine and screening for all 14 HPV types might result in a suboptimal balance of harms and benefits" [21]. Also, a Swedish study found proof that a considerable majority (85.3%) of the screen-detected cervical cancers were associated with HPV 16, 18, 31, 33, 45, or 52. The inclusion of the remaining eight HPV types covered by most screening tests only marginally increased the prevalence by 1.5% [22]. Our study further supports these observations, as we found that the eight additional types were only detected in 1.4% of the screen-detected cancers, while 93.0% (66/71) were caused by the HPV types 16, 18, 31, 33, and 45.

In Europe, reported statistics inform that most cervical cancer cases are caused by five high-risk HPV types (e.g., HPV 16, 18, 31, 33, and 45) [16]. Focusing on only these specific types in our material, 93.0% (66/71) of the screen-detected cancers and 80.4% (86/107) of the incidents among women with symptoms were accounted for. Interestingly, HPV 35 exhibited a more prominent presence among symptom-detected cancers compared to screen-detected cases, contributing to 6.5% of the former and 1.4% of the latter. Globally, HPV 35 has been identified as the eighth most prevalent type of cervical cancer, ranking seventh in Europe [16]. In our material, HPV type 35 appeared to have a higher association with cervical cancer development than HPV type 31 and, arguably, more significant than HPV types 52 and 58, being identified in only 1.2% of the symptom-detected cancers. The distinct presence of HPV type 35 raises considerations for its potential inclusion in screening tests and HPV vaccines. However, it is noteworthy that only one out of the eight women with HPV type 35-induced cancer was detected through screening, while the remaining were detected among women with symptoms. Further research is warranted to explore the clinical significance and optimal strategies for the detection and prevention of HPV type 35-related cervical cancers.

Cervical cancer is a significant health concern, particularly among young women under the age of 45, ranking as the second most common cancer type following breast cancer [23]. Recent data by Gravdal et al. indicate a threefold increase in cervical cancer incidence among Norwegian women under 30 years of age since the 1950s [24]. A large population-based study conducted in Sweden examined the age-specific incidence of

Invasive Cervical Cancer (ICC) categorized by HPV type, covering 2850 confirmed ICC cases from 2002 to 2011. The study identified two age groups with higher incidence rates: one in the 30–45 age range and another in older age groups, approximately 70–80 years [25]. Consistent with our study, the most common HPV types found in cervical cancer cases were HPV 16, 18, and 45, predominantly observed in women aged 35–40. Conversely, other oncogenic HPV types, such as 31, 33, 35, 39, 52, 58, and 73, exhibited a higher prevalence in older age groups. Two cross-sectional studies investigating HPV distribution in European ICC cases also reported a low age association for HPV 16, 18, and 45-positive cervical cancer cases [26], supporting our findings where the mean age was approximately 44 to 45 years for these three types. These findings underscore the importance of including HPV 45 in primary prevention strategies targeting HPV-related cervical diseases.

Whilst most cervical cancer research has been utilizing HPV DNA-based technology for prevalence and type distribution evaluations, our study adds to the existing gap in the literature on how HPV mRNA tests perform in cervical cancer tissue samples. The risk of degradation has been addressed as a limitation for RNA-based testing; however, among the 181 archived FFPE tissue samples in our material, only one sample was excluded because of a negative mRNA intrinsic sample control. The reported 91.1% positive agreement between HPV types detected by DNA and mRNA confirms mRNA to be stable in FFPE tissue samples for up to 23 years of storage. Analyzing the observed frequency of each of the seven genotypes included in both tests, we found that DNA testing resulted in 89 HPV 16, 31 HPV 18, and 13 HPV 45 infections, while mRNA testing reported 90, 32, and 13, respectively. HPV 33, 31, 52, and 58 were detected by DNA in 9, 4, 1, and 1 cases, while mRNA was detected in 8, 4, 1, and 1 cases. Similar results were presented by Rad et al., comparing HPV DNA to mRNA testing in cervical cancer material from South African women, concluding that a mRNA test could be a valuable tool to describe HPV type distribution in cervical cancer tissue [27].

Concerns have been raised that continuing broad HPV primary screening in an HPV-vaccinated population, where a significant proportion of individuals have already received protection against the most oncogenic HPV types, will yield reduced predictive values because of low HPV-related disease prevalence. Presumably, the unchanged prevalence of the non-targeted HPV types with limited or no oncogenic potential will outnumber the true positive HPV results [28–30]. The presented data aligns with previous research indicating that primary HPV screening, specifically focusing on the genotypes covered by the nonavalent vaccine, might enhance the effectiveness of screening programs in vaccinated populations [21,28,31]. By tailoring screening strategies targeting only the HPV types with the highest oncogenic potential, improved detection of cervical cancer and targeted interventions for at-risk individuals might be achieved.

The emergence of a 7-type HPV mRNA test, such as the PreTect HPV-Proofer'7, might present a promising advancement in the field of cervical cancer screening. Previous studies have indicated that a 7-type HPV mRNA test may offer improved performance when compared to a 14-type HPV DNA test in terms of balancing the benefits and harms of screening [32–35]. By targeting a restricted number of high-risk genotypes, the mRNA test reduces the likelihood of detecting transient or clinically insignificant HPV infections, thereby minimizing unnecessary follow-up procedures and potential psychological distress for patients. This targeted approach enables a more efficient and cost-effective screening process without compromising the overall effectiveness of cervical cancer detection. As such, the 7-type HPV mRNA test holds great promise as a valuable tool in cervical cancer prevention and control, particularly in populations where HPV vaccination has been implemented.

Fundamental questions have been raised regarding the possible HPV-type replacement and how this might impact type distribution and associated risk of cervical dysplasia in the post-vaccination era. Theoretically, when the HPV types targeted by the vaccine are reduced or eliminated, other HPV types may increase in prevalence and potentially contribute to cervical cancer cases. This concern arises from the concept that HPV types not

covered by the vaccine may fill the ecological niche left by the targeted types [36]. However, numerous population-based studies and real-world observations in countries with high HPV vaccine coverage have consistently shown a substantial decrease in the prevalence of HPV types included in the vaccine. Additionally, the incidence of cervical precancerous lesions and cervical cancer has significantly declined in vaccinated populations. While ongoing surveillance and research are important to monitor any changes in HPV type distribution, current evidence does not support the notion that type replacement negates the overall effectiveness of HPV vaccination. The widespread use of the HPV vaccine has proven to be a crucial strategy in preventing cervical cancer and improving public health outcomes [37–44].

*Strengths and Limitations*

The present study possesses several strengths that contribute to the reliability and validity of the findings. Firstly, as a population-based study, it includes the enrollment of all women diagnosed with cervical cancer over a 20-year period in Troms and Finnmark counties, Northern Norway. This comprehensive inclusion ensures that the study captures a diverse representation of women with cervical cancer within the region. Additionally, the utilization of the Norwegian Cancer Register enhances the accuracy and credibility of the patient's medical history and cancer diagnosis, safeguarding the integrity of the data. The inclusion of a second histopathology review to confirm the cancer diagnosis prior to enrollment further strengthens the study's validity by ensuring the accuracy of the cases included in the analysis. Moreover, the confirmation of the causative HPV genotype through both DNA and mRNA technologies adds robustness to the findings, bolstering the reliability of the HPV detection results.

Despite these strengths, there are several limitations to acknowledge. Firstly, the study's sample size may be considered relatively small when compared to larger national or international studies. However, within a regional context, the sample size surpasses the average for published studies, providing valuable insights into the specific population under investigation. Another limitation is related to the restricted number of HPV types included in the HPV mRNA test used in the study. This limited scope may result in the underrepresentation of other HPV genotypes not covered by the test, potentially impacting the overall distribution of HPV genotypes in cervical cancer.

Further, our study focused on documenting the HPV types present in women with cervical cancer prior to the introduction of the HPV vaccine. As a result, we cannot provide direct data on the occurrence or potential impact of type replacement following vaccination. Type replacement refers to the possibility of other HPV types increasing in prevalence and potentially contributing to cervical cancer cases because of the reduction or elimination of vaccine-targeted types. However, it is important to note that extensive research conducted in vaccinated populations has not shown evidence of significant type replacement to date [37–44]. Nonetheless, future studies specifically designed to investigate type replacement dynamics in the post-vaccine era would be valuable to better understand the long-term effects of HPV vaccination.

Lastly, the study also included women with HPV-negative formalin-fixed paraffin-embedded (FFPE) tissue samples that were not previously tested for HPV using liquid-based cytology (LBC) samples. Notwithstanding these limitations, the strengths of the study, including its population-based design, utilization of the Norwegian Cancer Register, rigorous diagnostic confirmation, and confirmation of HPV genotypes, contribute to the robustness and reliability of the findings.

## 5. Conclusions

Our findings contributed to the growing body of evidence supporting the importance of HPV testing and vaccination, providing valuable insights into the prevalence and natural distribution of HPV genotypes in non-vaccinated women diagnosed with cervical cancer within the specific region. The high percentage of women testing positive for HPV (92.7%)

highlights the significant role of this virus in the development of cervical cancer and emphasizes the potential impact of HPV vaccination in preventing HPV-related cervical cancer. The bivalent, quadrivalent, and nonavalent HPV vaccines showed promising efficacy in reducing the burden of HPV-positive cervical cancer cases. Importantly, by targeting the most prevalent HPV genotypes identified in cervical cancer cases, the nonavalent vaccine has the potential to prevent most cervical cancer cases in our region. HPV screening tests may be more effective if they focus on the HPV types included in the nonavalent vaccine. Supported by the high concordance observed between HPV DNA and mRNA test results in women with cervical cancer, the 7-type HPV mRNA test might be an advantageous strategy toward the ultimate goal of eliminating HPV-related cervical cancer as a public health burden.

**Author Contributions:** Conceptualization, S.W.S. and B.M.F.; methodology, S.W.S. and B.M.F.; formal analysis, S.W.S.; investigation, S.W.S., M.A. and B.M.F.; resources, S.W.S., M.A. and B.M.F.; data curation, S.W.S.; writing—original draft preparation, S.W.S. and B.M.F.; writing—review and editing, S.W.S., M.A. and B.M.F.; visualization, S.W.S. and B.M.F.; supervision, S.W.S.; project administration, S.W.S. and B.M.F.; funding acquisition, S.W.S. and B.M.F. All authors have read and agreed to the published version of the manuscript.

**Funding:** This research received no external funding. HPV testing was provided FOC by PreTect AS.

**Institutional Review Board Statement:** The study was conducted in accordance with the Declaration of Helsinki and approved by the Regional Committee for Medical and Health Research Ethics (REC North, 203384, 18 December 2020) for studies involving humans.

**Informed Consent Statement:** Patient consent was waived due to Norwegian regulations, which exempt quality assurance studies from written informed consent.

**Data Availability Statement:** The data presented in this study are available on request from the corresponding author. The data are not publicly available due to ethical restrictions.

**Acknowledgments:** We would like to extend our gratitude to the staff at the Department of Clinical Pathology at the University Hospital of North Norway for the histopathology evaluation and to all the laboratory staff at PreTect AS, performing HPV DNA/mRNA testing, for their great work and collaboration during this study.

**Conflicts of Interest:** S.W.S and M.A. declare no conflict of interest. B.M.F is an employee of PreTect AS.

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
