# Peer review of "Distribution of HPV Types in Tumor Tissue from Non-Vaccinated Women with Cervical Cancer in Norway"

_jmp, doi:10.3390/jmp4030015_

Round 1

Reviewer 1 Report

The manuscript is well written and was very friendly to follow. There, however, a few serious concerns as to the conclusions and recommendations made in the manuscript.

1.     Since there is a vaccine now aimed at a few oncogenic types, there are concerns that virus may find a way to be infectious and carcinogenic by type-replacement.  Any surveillance studies should consider how the type replacement would be identified, and the assay can find this. 

2.     We are now in the post vaccine era (since 2008), but the samples in this study include both pre and post vaccine (1995-2015). This raises serious concerns as to the fitness (whether the finding fit the pre or post vaccine period) of the observed type distribution.

3.     The authors state that the 7 type RNA test is equivalent to 45 type DNA test from the k-value (0.61). While this may be interpreted as good agreement, the McNemar p-value would indicate the results are quite different significantly based on discordant. Further, sample size is too small and type 35 and 39 are not included in the RNA test.

4.     Types to be included in screening/surveillance studies should consider those types that are likely to occupy the new “niche”, and all members of alpha 9 and 7 may have increased probability of being replaced. The authors should discuss this scenario when they raise points in favor of including only type 16, 18, 31, 33 and 45.

Author Response

Reviewer 1: 
The manuscript is well written and was very friendly to follow. There, however, a few serious concerns as to the conclusions and recommendations made in the manuscript.

1.     Since there is a vaccine now aimed at a few oncogenic types, there are concerns that virus may find a way to be infectious and carcinogenic by type-replacement.  Any surveillance studies should consider how the type replacement would be identified, and the assay can find this. 

Our response: Thank you for your valuable feedback on our manuscript regarding HPV types in women with cervical cancer. We acknowledge the need to include the scientific discussions regarding type replacement in the context of HPV vaccination to our manuscript, and the revised manuscript has been updated accordingly (Lines 341-355).

2.     We are now in the post vaccine era (since 2008), but the samples in this study include both pre and post vaccine (1995-2015). This raises serious concerns as to the fitness (whether the finding fit the pre or post vaccine period) of the observed type distribution. 

Our response: Thank you for raising your concern regarding the fitness of the observed type distribution in our study, considering the pre- and post-vaccine era. We fully acknowledge the need to clarify this important aspect, and have updated the manuscript with more details to the study design section and with a clear reference throughout the manuscripts that reported data was from the pre-vaccine era. 

In detail, the revised manuscript has been updated with a new title: “Distribution of HPV Types in Tumour Tissue from Non-Vaccinated Women with Cervical Cancer in Norway”, the abstract address “genotype prevalence in the pre-vaccine era” (L12) from “non-vaccinated women” (L16).  

Introduction (L87-92) informs that our study specifically focuses on the pre-vaccine period, aiming to contribute to the understanding of the natural distribution of HPV in cervical cancer cases, that might serve as a relevant baseline for evaluating the impact of vaccination on HPV type distribution in future studies post-vaccination. Study population (L109-113) has been made more clear describing the women`s not vaccinated status and year of birth. In Norway, the HPV vaccine was first introduced in 2009 for women born in 1997 and later. A catch-up vaccination campaign was implemented in 2016 for women born between 1991 and 1996. However, it is crucial to note that there has been limited vaccination of women born in other birth cohorts in Norway. None of the women in the study population had received the HPV vaccine, nor born in cohorts that received the vaccine. 

We acknowledge that the introduction of the HPV vaccine could potentially influence the distribution of HPV types in cervical cancer cases, and that reported data only reflect the HPV type distribution in cervical cancer cases before vaccination was implemented. Hence, the Strengths and Limitations have been updated (L378-387), addressing the limitation that the study data originated from the pre-vaccine era, which cannot provide data on the occurrence or possibility of type-replacement post-vaccination. Future studies specifically designed to investigate type replacement dynamics in the post-vaccine era would be valuable to better understand the long-term effects of HPV vaccination. 

3.     The authors state that the 7 type RNA test is equivalent to 45 type DNA test from the k-value (0.61). While this may be interpreted as good agreement, the McNemar p-value would indicate the results are quite different significantly based on discordant. Further, sample size is too small and type 35 and 39 are not included in the RNA test. 

Our response: Thank you for your comments regarding the comparison between the 45-type HPV DNA test and the 7-type HPV mRNA test used in our study. We appreciate the opportunity to address your concerns and provide further clarification. 

You mentioned that the k-value of 0.617 indicates good agreement between the two tests, but the McNemar p-value suggests significant differences based on discordant results. We understand your point, and it is important to acknowledge that there may be variations between the two tests. The revised manuscript includes a reference to the McNemar test (L169-176). However, it is crucial to consider the context and purpose of each test in our study. 

The primary objective of our study was to assess the prevalence of HPV types in cervical cancer cases using two different tests: the 45-type HPV DNA test and the 7-type HPV mRNA test. The focus was on evaluating the potential efficacy of HPV vaccines in preventing cervical cancer, particularly the 9-valent HPV vaccine, which targets the seven HPV types covered by the mRNA test, and further to demonstrate how the HPV mRNA test performed in cervical cancer tissue samples. 

Regarding your comment on sample size, this limitation has been clearly addressed in the "strengths and limitations" section (L370-374). We acknowledge that it is relatively small, as with many studies involving rare diseases such as cervical cancer. However, our study included a comprehensive analysis of 178 valid cases, providing valuable insights into the distribution of HPV types in cervical cancer within the selected population. 

We understand your concerns about the non-inclusion of HPV 35 and HPV 39 in the mRNA test, if changing screening practices in the post-vaccination era from a 14-type HPV test to a test targeting only the vaccine types. While these types may be more prevalent in cervical cancer cases in specific populations, our study focused on evaluating the potential efficacy of the 9-valent HPV vaccine, which targets the seven high-risk HPV types included in the mRNA test. As discussed within the manuscript (L274-286), HPV 35 exhibited a more prominent presence among symptom-detected cancers compared to screen-detected cases, and further research is warranted to explore the clinical significance and optimal strategies for the detection and prevention of HPV type 35-related cervical cancers. 

4.     Types to be included in screening/surveillance studies should consider those types that are likely to occupy the new “niche”, and all members of alpha 9 and 7 may have increased probability of being replaced. The authors should discuss this scenario when they raise points in favor of including only type 16, 18, 31, 33 and 45. 

Our response: Thank you for your additional comments and concerns regarding type replacement in screening/surveillance studies. We appreciate the opportunity to address this topic and provide further clarification how this has been discussed in our manuscript and what updates has been made to the revised document.  

As explained in our response to your two first comments, we focused on evaluating the distribution of HPV types in cervical cancer cases before the introduction of HPV vaccination in Norway. While our study specifically examined the prevalence of HPV types 16, 18, 31, 33, 45, 52 and 58 it is important to consider that the 9-valent HPV vaccine, which includes these seven types, has demonstrated high efficacy in preventing cervical cancer and related precancerous lesions. Nevertheless, we acknowledge that ongoing surveillance is essential to monitor HPV type distribution and potential changes over time. Continued research and monitoring efforts can help identify any shifts or emerging HPV types that may require consideration in future screening/surveillance studies, and this aspect has been included in the revised manuskript (L341-355), (L378-387). 

While discussing the number of types to be included in a screening test postvaccination (L260-270), (L318-340), we addressed the five types as the most common types identified in European Cervical cancers, while arguing the importance of the seven types included in the vaccine, as a favorable approach for future screening tests. 

We sincerely thank you for taking the time to review our manuscript and for your constructive feedback. If you have any additional questions or require further information, please do not hesitate to let us know. We value your insights and appreciate your contributions to the advancement of our research. 

Reviewer 2 Report

This study has been investigated the prevalence of HPV genotypes in 178 cases of cervical cancer in order to evaluate the effectiveness of HPV screening and vaccination in reducing the incidence of cervical cancer. The results were compared with an HPV type 7 mRNA test (16,18,31,33,45,52,58) to identify the causal genotype. Concluding that the nonavalent vaccine could prevent 93.3% (154/165) of cases of cervical cancer.

I find this study, which I have read with pleasure, very interesting for readers. The work is well written, the results are clear and the conclusions are supported by the results.

Author Response

Reviewer 2 
This study has been investigated the prevalence of HPV genotypes in 178 cases of cervical cancer in order to evaluate the effectiveness of HPV screening and vaccination in reducing the incidence of cervical cancer. The results were compared with an HPV type 7 mRNA test (16,18,31,33,45,52,58) to identify the causal genotype. Concluding that the nonavalent vaccine could prevent 93.3% (154/165) of cases of cervical cancer.  

I find this study, which I have read with pleasure, very interesting for readers. The work is well written, the results are clear and the conclusions are supported by the results.  

Our response: Thank you for the encouraging comments to the value of our manuscript. We greatly appreciate your thorough review and the overall positive assessment of our work. We sincerely thank you for taking the time to review our manuscript. If you have any additional questions or require further information, please do not hesitate to let us know. We value your insights and appreciate your contributions to the advancement of our research.

Round 2

Reviewer 1 Report

1. line 175, is it 14 or 45 type test?

2. McNemar test added in the methods but no mention of it in the results regarding outcome of the test. Consider including how the test performed

Author Response

Response to Correction 1:

Reviewer 1: line 175, is it 14 or 45 type test?

Our response: In our daily routine, we use a 14-type HPV DNA test (Roche Cobas 4800). However, in this study, we utilized a 45-type HPV DNA test, specifically the Reverse Line Blot (RLB) assay. We apologize for any confusion, and the paragraph "Statistical Analyses" has been updated accordingly:

"Statistical Analysis: Data were analysed using Statistical Package for Social Sciences (SPSS) v28 (IBM). For comparisons between the 45-type HPV DNA test and the 7-type HPV mRNA test, we used the McNemar test with continuity correction, which is appropriate for analysing paired nominal data in a 2 x 2 table."

Response to Correction 2:

Reviewer 1: McNemar test added in the methods but no mention of it in the results regarding the outcome of the test. Consider including how the test performed.

Our Response: Thank you for bringing this to our attention. We have now included the results of the McNemar test in the Results section:

"The overall concordance rate for the two different HPV-tests was 91.0% (162/178), with a kappa value of 0.617 and P-value <0.001. The McNemar chi-squared statistic was equal to 5.06 (P=0.024), see Table 2."